# HIV self-testing and partner notification strategies for key populations in low- to upper-middle-income countries: A mixed-methods systematic review

Lekey Khandu [1]*, Jonathan Hallett [1] Gemma Crawford[1], Justine E. Leavy[1], Daniel Vujcich[2]

**1** Faculty of Health Sciences, School of Population Health, Curtin University, Perth, Western Australia, Australia, **2** Western Australian AIDS Council, Perth, Western Australia, Australia

* lekey.khandu@postgrad.curtin.edu.au

## Abstract

### Background

HIV self-testing (HIVST) enhances case diagnosis, but information on its integration for index testing to partner notification is limited. Assessing the acceptability and feasibility of index HIVST and partner testing among people living with HIV (PLHIV) and undiagnosed key populations is critical to ending the AIDS epidemic.

### Methods

A mixed-methods systematic review using a convergent segregated approach was conducted using the Joanna Briggs Institute's methodology. Four databases were used to conduct a literature review from October 2023 to March 2024, which included studies published between 2016 and 2023. Rayyan software was used for full-text screening. Meta-analysis was deemed infeasible; however, a qualitative meta-aggregation approach was conducted.

### Results

A total of 4076 studies were retrieved, and 76 studies met the inclusion criteria after a full review. Most of these studies were from the African region, with only one from South Asia and a few from East Asia. Index HIVST and partner testing were found acceptable and feasible among PLHIV and key populations. Despite low partner elicitation ratios through assisted partner notification, a higher positivity rate was noted among notified individuals. Preferences for index HIVST and partner testing varied, with more inclination for assisted and passive partner referrals to overcome the risk of HIV status disclosure. Assisted partner notification (aPN) showed a low cost per infection averted, indicating it's a cost-effective intervention. Available evidence was

**Data availability statement:** All relevant data are within the manuscript and its Supporting Information files.

**Funding:** This study was financially supported by Curtin University of Technology in the form of a PhD scholarship award received by LK. No additional external funding was received for this study.

**Competing interests:** The authors have declared that no competing interests exist.

skewed towards married couples, with less evidence on unmarried and undiagnosed key populations.

## Conclusion

The index HIVST was found to be acceptable and feasible in reaching index partners, including untested and undiagnosed partners of key populations, when various testing approaches are used. Understanding the index HIVST among unmarried HIV index cases, and partner testing of undiagnosed key populations is important, particularly in the Southeast Asian region, to bridge the current HIV case detection gaps. Although no specific cost-effectiveness studies were found for the index HIVST, the aPN was found to be cost-effective.

## Study registration

PROSPERO Number: CRD42023475417 dated 2023. Available from: https://www.crd.york.ac.uk/prospero/display_record.php?ID=CRD42023475417.

## Introduction

HIV testing is vital for timely diagnosis, linkages to care, and treatment [1]. However, access to HIV testing was prevented due to stigma and discrimination, particularly among sexual and gender minorities [2]. HIV self-testing (HIVST) enhances testing access, especially for key populations, helping to bridge the case detection gap [3]. It enables key populations to test privately without visiting health centres [4]. Structural, social, economic, political, and environmental factors inhibit HIV testing uptake [5]. Promising uptake of HIVST has been observed among key populations such as men who have sex with men (MSM), transgender (TG) persons, female sex workers (FSWs), people who inject drugs (PWID), and first-time testers [6]. Community-based testing (CBT) approaches increase HIV testing uptake among key populations compared to healthcare provider-based testing services [7–9]. As HIV incidence decreases, partner notification becomes crucial for identifying new cases among contacts of newly diagnosed PLHIV [10]. Evidence shows intensified testing strategies, including HIVST as an alternative testing option [11], but concerns exist regarding poor or absent pre-test counselling, which poses challenges in confirming results and linking to the continuum of care [12,13].

Given these developments, integrating HIVST into partner notification strategies represents a promising yet underexplored opportunity to enhance testing coverage and linkage to care. Building on the growing role of HIVST among key populations, it is essential to explore how HIVST can be integrated into partner notification strategies through index testing approaches. Recognizing evidence-based methods to use the index HIVST for partner notification is important for early detection and timely linkage to treatment. Index testing is a voluntary consultative approach between a trained provider and a diagnosed person living with HIV (PLHIV) (the index case) to

identify/elicit exposed contacts (sexual partners, injecting drug users, and biological children) and offer targeted HIV testing [10,14]. The World Health Organization (WHO) defines partner notification as an integral part of index testing involving the disclosure of the HIV status of index cases to their contacts, through passive or assisted partner notification (aPN) [10]. Passive partner referral (pPR) involves index cases self-disclosing their status without provider mediation, and aPN involves active participation from providers and may take three forms: *contact referral* involves requiring the index PLHIV disclose their HIV status and refer partners for testing within a specified timeframe; *provider referral* involves trained providers directly contacting partners for testing with the index case's consent; *dual referral* involves both index cases and providers working together to notify partners and offer testing [10].

Despite partner notification's long history in HIV management, integrating index HIVST within partner notification services (PNS) may strengthen efforts to achieve the global 95-95-95 targets to end the HIV/AIDS epidemic. Evidence shows a high likelihood of HIV infection among sexual and drug-injecting partners of diagnosed PLHIV [15,16]. Traditional provider-based index testing has proven useful for case detection and the timely initiation of treatment [17,18]. Although many low and middle-income countries (LMICs) are scaling up HIVST, integration into PN remains inconsistent and limited [10,19]. However, even with the evidence supporting HIVST effectiveness, there is limited understanding about how HIVST is integrated into partner notification services in lower to upper-middle-income countries, particularly among key populations. Moreover, to date, there is no mixed-methods systematic review that has comprehensively covered strategic information on both the index HIVST and traditional facility-based PNS. Therefore, this mixed-methods systematic review aims to assess the acceptability and feasibility of index HIVST and other partner notification strategies for reaching sexual and drug-injecting partners of index cases and other undiagnosed key populations in low- to upper-middle-income countries. This review identifies and synthesizes the literature to answer the following questions:

1. What is the acceptability, feasibility, and level of uptake of HIVST for partner notification compared to facility-based testing?

2. What is the effectiveness of aPN and other partner notification strategies with and without HIVST, in increasing partner testing, HIV case identification, and linkages to care among index cases and undiagnosed key populations?

3. What factors, including barriers and facilitators, influence the feasibility, safety, and implementation of HIVST within partner notification services, among index cases and undiagnosed key populations?

## Methods

This review followed the JBI methodology for mixed-methods systematic reviews [20]. The review was conducted per the Preferred Reporting Items for Systematic Reviews and Meta-analyses (PRISMA) guidelines [21]. The completed PRISMA checklist is presented as a supplementary S1 Table. This review was registered prospectively with PROSPERO (CRD42023475417) [22] to ensure rigor and transparency. Other than a small change to the wording of the second research question to more accurately reflect that effectiveness is a comparative concept, no major amendments were made to the information provided at registration or in the protocol. No ethical approval was required. This mixed-methods systematic review aims to utilize quantitative findings to assess the uptake and effectiveness, while qualitative studies identify barriers, facilitators, and contextual details.

### Eligibility criteria

An initial search in PROSPERO, Google Scholar, Web of Science, PubMed databases and the Joanna Briggs Institute (JBI) Evidence Synthesis identified no current or progress systematic reviews on this topic. While prior reviews explored general HIVST acceptability, and few have assessed HIVST and aPN separately [3,23], none have systematically compared HIVST and aPN as index testing strategies for partner notification using a mixed-methods approach across diverse populations in

low- to upper-middle-income countries. This review considered peer-reviewed articles from quantitative, qualitative, and mixed-methods studies from low to upper-middle-income countries, classified by the World Bank GNI per capita, Atlas method [24]. Studies published in English between 1 January 2016 and 30 December 2023 were included to align with WHO guidelines on HIVST and partner notification [10]. Quantitative studies were included that identified the level of uptake of index HIVST versus facility-based testing, the effectiveness of aPN for partner notification among diagnosed PLHIV and undiagnosed key populations, and the cost-effectiveness of aPN for index testing. Both the quantitative and qualitative components examined barriers, facilitators, and social harms associated with index testing for PNS. Furthermore, qualitative and quantitative data from mixed-methods studies were extracted and analyzed separately following the JBI convergent segregated approach.

## Data outcome definition

Aligned with the review questions, the outcomes were operationally defined according to the existing literature [25]. Acceptability was defined as the proportion of individuals who self-tested among those offered testing. At the same time, uptake was computed as acceptability and the actual HIVST-use percentage. Feasibility was defined as the completion of HIVST and counselling as a result of ease of use, correct interpretation of results, and links to care. The effectiveness is a key outcome of partner notification, including partner testing uptake, new diagnosis and linkage to care among the index and their partners.

## Search strategy

The formulated search terms for HIV, testing, and partner notification concepts were explored for synonyms and subject headings, with the full search strategy presented in the supplementary S2 Table. The Web of Science, Embase, Global Health, and Medline databases were used to search for qualitative and quantitative data from 1 October 2023 to 30 March 2024. The first 300 results from Google Scholar were also assessed for inclusion [26]. Boolean terms, truncations, and medical subject headings (MeSH) were used where relevant, except in Google Scholar. Reference lists of included studies were also reviewed to identify additional relevant articles.

## Study selection

All identified studies were collated and exported into EndNote. [27] for duplication removal and then imported into Rayyan [28] for detailed text review. After a pilot test, the first author (LK) screened titles and abstracts. The first author conducted a full-text review for all studies, and the second review was distributed randomly to other research team members. Disagreements were resolved through group discussion.

## Methodological quality appraisal

The MetaQAT [29] tool was applied as previously adapted by Leavy and colleagues [30] to critically evaluate the selected studies with minor modifications for clarity. MetaQAT allows rigorous appraisal of multiple study designs and non-standard domain-like applicability [29,31]. The First author assessed all selected studies for methodological quality. Ten per cent of articles were randomly selected and evaluated by two other authors, independently. Methodological quality was assessed using questions under each domain (relevance, reliability, validity, and applicability). Each study's risk of bias (ROB) was assessed using the MetaQAT tool, scoring nine criteria from 0 to 2 (maximum 18). Studies were classified as low (15–18), moderate (11–14), or high risk of bias (≤10), consistent with similar reviews [30]. ROB ratings guided sensitivity analyses and evidence interpretation.

## Data extraction/collection

Key data from the included studies were extracted using Rayyan software and organised into a charting table by the first author. The charting table, developed using Microsoft Excel, aligned with the research question and PRISMA guidelines

[21]. The charting table was pilot-tested to assess its feasibility and reliability before implementation. The data table captured the study characteristics and the phenomenon of interest.

## Data synthesis and integration

Following the JBI convergent segregated approach [32], qualitative data were synthesized independently before integration. Quantitative data were synthesized narratively using descriptive and inferential statistics, without meta-analysis due to heterogeneity. Sources of heterogeneity included differences in study design, population groups (e.g., PLHIV, MSM, FSW, PWID), HIVST distribution strategies (e.g., aPN, partner-provided referral), measured outcomes (e.g., uptake, positivity rate, linkage), and geographic or health system contexts across low- to upper-middle-income countries.

A meta-aggregation approach was used for qualitative data synthesis [33]. This approach aims to synthesize qualitative data to provide an overview of existing evidence, identify evidence gaps and make recommendations instead of re-interpreting the original findings of the included studies [34,35]. The three levels of meta-aggregative evidence used were: unequivocal, equivocal, and unsupported, as shown in Table 1. Common themes, patterns and concepts were identified using Excel and manually coded to generate relevant categories for meta-aggregation [34]. Findings were interpreted across service delivery and implementation factors influencing the uptake of HIVST and aPN across different populations, relationship types and settings. Appropriate comparisons were made to understand the effectiveness and feasibility of each partner notification model in terms of technical efficiency and cost, with the aim of future adaptation to scale up HIV testing and timely care linkages among key populations. The qualitative meta-aggregation synthesized findings are shown in S3 Table. The findings related to subgroup populations (MSM, FSW, PWID, transgender persons, and index clients) were analyzed narratively to identify characteristic differences, while the regional context was considered to understand the equity-related differences in accessibility and feasibility.

## Results

### Study inclusion

A total of 4076 studies were retrieved and exported to EndNote (Fig 1). After removing duplicates, the titles and abstracts were reviewed and screened based on the inclusion criteria. A total of 2791 studies were excluded, resulting in 213 articles for full-text review. After a full-text review, 76 articles that met the inclusion criteria were considered for review. Inclusion and exclusion reasons for all full-text articles are detailed in Supplementary S4 Table.

### Study characteristics

Seventy-six studies were included: 54 quantitative, 15 qualitative, and seven mixed-method studies published from 2016 to 2023, as presented in supplementary S5 Table. Quantitative designs included cross-sectional (n = 23), observational cohorts (n = 16), randomized control trials (RCTs) (n = 18), cost-effectiveness(n = 4), quasi-experimental (n = 3), and case-control (n = 1). Studies were conducted in eight low-income countries, eight lower-middle-income countries, and four upper-middle-income countries.

The largest number of publications came from Kenya (n = 20) [36–52,53,54], followed by China [55–61], Nigeria [62–68], and Malawi [69–74] with seven studies each. Indonesia published (n = 5) studies [75–79]. While Uganda [80–83]

**Table 1. Definition of the meta-aggregative level of evidence.**

| Level of evidence | Definitions |
| --- | --- |
| Unequivocal | The findings are accompanied by an illustration that is beyond a reasonable doubt and therefore not open to challenge. |
| Equivocal | The findings are accompanied by an illustration lacking a clear association with it and, therefore, open to challenge. |
| Not supported | The findings are not supported by the data. |

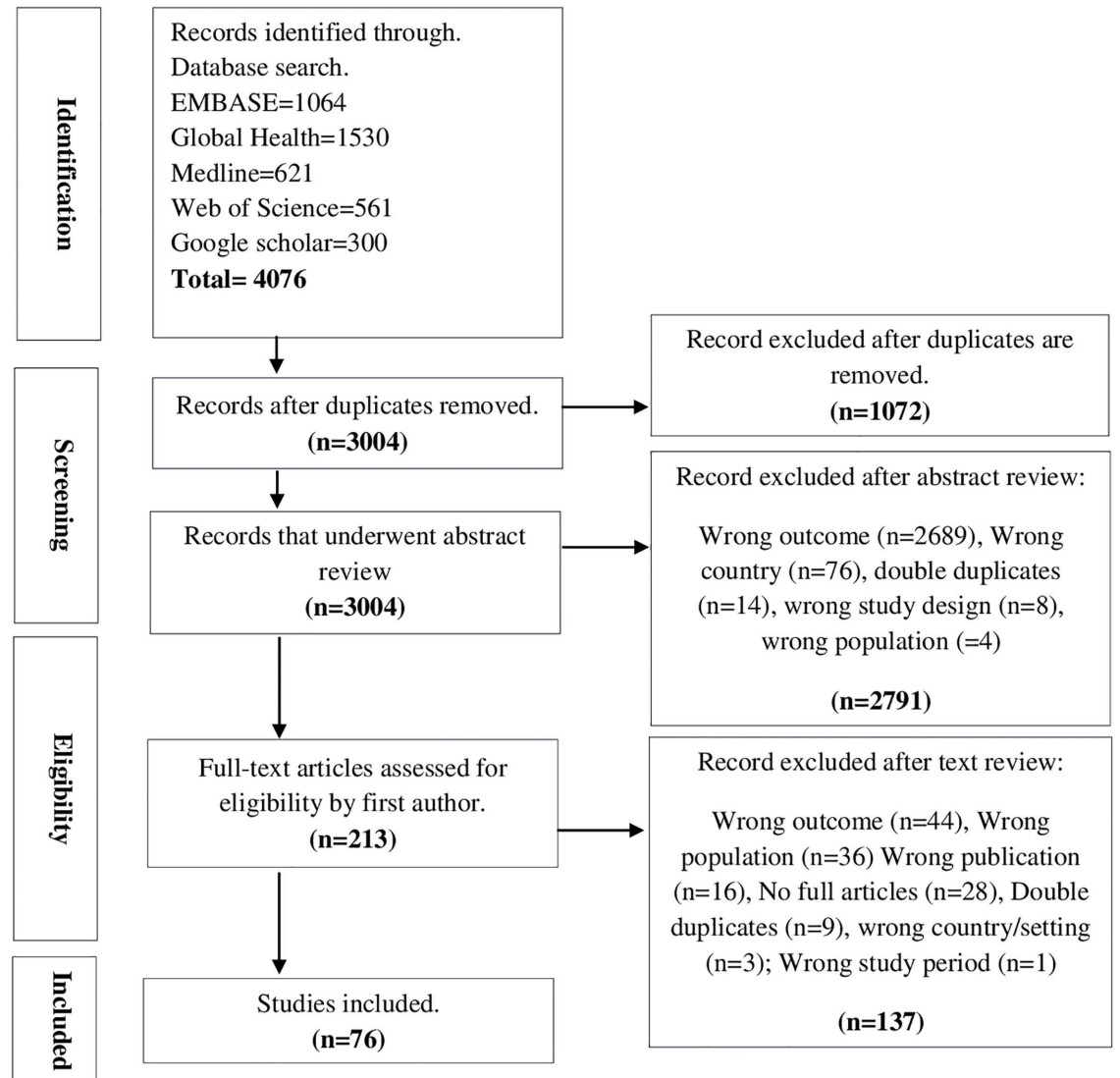

**Fig 1. PRISMA diagram.**

and Zambia [84–87] published four each. Mozambique [88–90], Rwanda [91–93] and South Africa [94–96] published three studies each. Three countries published two studies each: Tanzania [97,98], Ethiopia [99,100], and Mexico [101,102]. Seven countries published one study each: Zimbabwe [103], Botswana [104], Cameroon [105], India [106], Vietnam [107], Mali [108], and Guinea-Bissau [109].

Out of 76 studies, 15 targeted key populations: seven focused on MSM [55–59,61], four on FSWs [36–38,85], three on MSM, FSWs and PWID [62,67,68], two on MSM and TG [101,102], and one on male prisoners [77]. While the remaining 61 studies focused on index clients and their sexual partners.

## Methodological quality and risk of bias

Seventy-six studies underwent methodological assessment using a tailored quality checklist adapted from MetaQAT [31]. As outlined in the methods, the tool evaluates four domains: relevance, reliability, validity, and applicability, with an

overall ROB score ranging from 0 to 18. Most studies (n = 72; 95%) were rated as low risk, indicating high methodological quality, with clearly defined objectives, appropriate study designs, and transparent reporting, which support the reliability and consistency of the findings for public health interpretation. A small number assessed as having moderate risk (n = 4; 5%). Studies with moderate risk typically exhibited limitations due to self-reporting or selection bias, and incomplete bias mitigation. Accordingly, all studies were retained for data extraction to support a comprehensive comparison of methods. Detailed quality appraisal and RoB results are presented in a supplementary S6 Table. Given the clinical and methodological heterogeneity across studies, including differences in populations, study designs, and HIVST distribution approaches, sensitivity analysis and statistical assessment of heterogeneity were not conducted.

## Quantitative findings

**Acceptability, feasibility and uptake of HIVST for partner notification compared with facility-based testing.** Out of 22 studies on HIVST for partner notification [36–40,55–57,61,69–71,73,74,83–85,94,95,97,107,108], eight provided data on the uptake of HIVST among index cases and key population partners [39,56,57,70,71,83,84,95]. The latter eight studies found a higher uptake of HIVST compared to facility-based testing, with the increases ranging from 14% to 70%. For example, a Malawian RCT among antiretroviral therapy (ART) cases showed 71% testing uptake in the index HIVST group and 25% in the partner referral slip (PRS) group (aRR 2.77, 95% CI [2.56 to 3.00], p ≤ 0.001) [70]. In Zambia, index HIVST uptake among the male partners of HIV-positive and negative Antenatal Care (ANC) women was 68% and 74%, respectively, compared to 4% and 8% in the facility-based testing group [84]. Another RCT showed a slight increase in testing uptake among the HIVST group compared to the facility-based testing group (HR 1.04, 95% CI: 0.79-1.37) [83]. In a Kenyan observational cohort study, the direct distribution of HIVST by index case resulted in 52% of partners being tested, compared to 38% in facility-based settings (AOR: 1.78, 95% CI: 1.13 - 2.78) [39]. A Malawian RCT among ANC women showed 89.3% HIVST uptake for male partners, 73% in HIVST with financial incentives and 35% in the facility-based arm [71]. An observational cohort study found 89.3% of FSWs' partners tested via direct distribution of HIVST, compared to 79.6% via coupon referral via health facility [85]. A Chinese cross-sectional study showed 87% HIVST uptake among MSM partners [56]. Another Chinese RCT among HIV-positive MSM depicted a higher uptake of aPN using HIVST than routine partner services (rPS) (48.1% vs. 21.9%, p < 0.001) [57].

**Effectiveness of aPN services.** Twenty-one studies [41,45,54,58,62–68,80,81,86–88,91,92,104,109,110] reported on the effectiveness of aPN in increasing partner testing uptake, new diagnoses and linkages to care. An observational study found that aPN increased partner notification by 13%, HIV testing by 101%, and new diagnoses by 126% among partners [88]. aPN also increased program reach to 93%, with significant case detection among identified index partners [81,87,104]. In Botswana, aPN reached 5,175 index cases (87%) in phase one and 1,265 index cases (95%) in phase two, with positivity rates of 22% and 31%, respectively [92]. A West African quasi-experimental study found 697 index cases referring 118 partners for aPN [109]. A Kenyan RCT revealed a significant increase in partner testing, new diagnoses, and linkages to care [41]. In Rwanda, index testing effectively identified HIV-positive individuals aged 50 +, particularly among female partners and those notified by newly diagnosed index cases [93]. A Malawi RCT found aPN increased partner clinic attendance more among women (PD 0.17) and previously diagnosed individuals (PD 0.20) than passive partner notification, with contract partner notification alone having similar effects [110]. Further, for participants with lifetime but not recent IPV, APS proved both effective and safe in Kenya [54].

In Nigeria, an observational study identified 3,119 index cases, which elicited 8,889 sexual partners, resulting in a high index partner ratio of 1:2.9, yielding a 79.4% testing uptake and a 49.7% positivity rate [62]. In Zambia, aPN elicited 845 partners from 604 index cases, resulting in a low index partner ratio of 1:1.4 [86]. In Nigeria, a low index partner ratio of 1:1.02 was also reported [63]. Uganda reported a 58% testing rate and a 32% positivity rate, with a 1:1.4 ratio of index partners [80]. Mozambique and Rwanda reported similar ratios of 1:1.2 [64,91]. In Kenya, aPN resulted in 77% testing with a 44% positivity rate among male partners [45]. In Nigeria, aPN delivered through provider and peer navigator

support among key populations led to high testing uptake (84.1%), a high positivity rate (49.7%), and 93% linkage to care, demonstrating strong effectiveness in community-led HTS settings [67]. Two Chinese studies showed index partner ratios (1:1.1 and 1: 0.92) with testing rates exceeding 99.6% [58,68], while three other studies with index partner ratios between 1:1–1:1.01 [64–66].

**Partner notification strategies with HIVST.** Several studies assessed the effectiveness of different distribution approaches for HIVST kits within the partner notification services. Six RCT studies [57,61,70,74,97,111] demonstrated that aPN effectively distributed HIVST kits to index partners, with rates ranging from 35% to 71%. For example, a Chinese RCT reported that aPN led to 35% of index cases having≥1 partner test (0.5 partners per index case) compared to 17% (0.2 partners per index case) in the passive partner notification arm; moreover, 49% of disclosed partners tested compared to 28% in the passive arm [61]. Similarly, another study from China showed that 65.9% of index clients had ≥ 1 partner tested, compared to 31.6% in regular partner notification; HR: 2.86 (95% CI: 2.03–4.03) [57]. A Malawian RCT by [70], index HIVST, although not labelled as aPN, functioned as a facilitated or assisted partner notification strategy, achieving 71% partner testing vs. 25% under passive referral (aRR: 2.77). Another RCT in Kenya reported 71% of male partners tested in the HIVST arm vs. 52% in the invitation arm. Furthermore, HIV-positive clients, 80.5%, were offered HIVST through assisted partner notification (aPN), resulting in a partner testing ratio of 0.56 per index client and a high HIV positivity rate of 41.9% among tested partners [107].

Seven other studies [39,56,74,83–85,111] have highlighted different HIVST distribution models, including ANC-based, peer-led, digital, and provider-assisted, with partner testing rates ranging from 28.2% to 94.9%, depending on the population and distribution strategy. Among ANC-based approaches, a Kenyan RCT among 600 antenatal/postpartum women showed that secondary distribution of HIVST achieved 90.8% partner testing versus 51.7% with invitation cards, and 75.4% versus 33.2% for couples testing [74]. A Ugandan RCT among pregnant women living with HIV found no significant difference in male partner testing or linkage to care between secondary distribution of HIVST and standard invitation-based partner referral [83]. In the peer-led model, a direct delivery of HIVST kits by peer educators in Zambia achieved the highest testing uptake among female sex workers (94.9% at 1 month, 84.1% at 4 months), compared to coupon-based collection (84.4%, 79.8%) and standard-of-care referrals (88.5%, 75.1%) [85].

Under the provider-assisted strategies, HIVST-supported pPR in Zambia led to significantly higher male partner self-testing uptake compared to standard contact referral: 68% vs. 4% in trial one, and 74% vs. 8% in trial two [84]. Furthermore, digital and information-based models also demonstrate effectiveness. A cross-sectional study in China found that information about aPN as a choice-based approach increased HIVST uptake nearly threefold compared to couples' HIV counselling and testing (AOR 2.9; 95% CI: 1.6, 9.3) [56]. Similarly, 52% of partners in the HIVST strategy tested versus 38% in the invitation strategy (AOR: 1.78, 95% CI: 1.13 to 2.78) [39]. A digital referral study in China found that 28.2% of MSM successfully referred partners for HIV testing, highlighting the potential of digital aPN to extend reach in key populations [111].

**Partner notification strategies without HIVST.** To contextualize the role of HIVST in partner notification, this review also examined partner notification methods used in index testing without using HIVST. Five studies [42,62,75,107,109] compared provider referral (aPN) with pPR, consistently showing higher uptake/acceptance for the provider referral-assisted model for index testing. For instance, among 7,556 key population index cases in Nigeria, 68.3% preferred provider referral, 30.1% pPR, 1.1% contact, and 0.5% dual (referral) [62]. While in a separate study, contact referrals led to successful notification of 67% of index partners versus 4% through pPR [42]. In Indonesia, provider referral had higher notification rates (53%) than pPR (18%) [75]. Similarly, Vietnam also reported 44.3% provider referral, 43.7% pPR, and 5.1% dual referral [105]. Another study found that 90% preferred pPR, 51% provider referral and 3% contact referral [109].

While the remaining six studies [41,64,72,80,87,91] highlighted varied partner notification preferences, including contact-based, self-disclosure, and dual referral strategies, which revealed more heterogeneous findings. In Mozambique, 94% accepted provider referrals [64], whereas in Zambia, preferences were 50% for the provider, 46% for the contact,

 

and 4% for dual (referral) [87]. Particularly, in Zambia, regardless of referral type, index testing effectively identified HIV-positive men [87]. In contrast, a study from China among FSW and MSM found that 54% preferred pPR, 22% contact referral, 19% provider referral and 5% dual referral [41]. In Uganda, 45.5% self-disclosed, 31.4% used provider referral, 19.5% assisted disclosure, and 3.6% unknown methods [80]. A study in Rwanda reported 44.1% provider, 33.9% contact, and 22.0% dual referral, with no correlations to case diagnosis [91]. High partner notification rates were observed across combination interventions compared to pPR [72].

**Barriers, facilitators, and social harms associated with HIVST for partner testing.** Ten quantitative studies reported a low prevalence of social harm, including relationship dissolution, intimate partner violence, verbal abuse, and loss of financial support, related to the index HIVST for partner notification [37,39,55–57,70,71,83,84,94,107]. However, three studies highlighted barriers to index HIVST [39,55,70]. In Malawi, 16% of the index partners struggled with HIVST instructions, 10% couldn't interpret the results, and three individuals didn't trust the results [70]. In China, 54.2% of index cases did not receive instructional videos, and 19.4% reported their partner's difficulties with HIVST [55]. Another study found that 94 out of 153 preferred oral HIVST over blood-based methods, citing ease and privacy as motivators [39].

Six studies highlighted the social implications of HIVST for partner testing [37,39,57,71,84,107]. Although IPV cases were rare, they were not negligible. In Zambia, 3/116 index cases reported intimate partner violence (IPV) [84]. In Malawi, three physical violence and four marital disputes were reported [71], while Vietnam reported two cases of suicidal thoughts without social harm [107]. A cohort study found that 14% experienced social harm, including relationship dissolution and emotional abuse [39]. Additional studies showed similar IPV prevalence. For index HIVST, reported rates ranged from 1.8% to 17.3%, and for facility-based testing, from 1.9% to 16.4% [37,57].

**Cost-effectiveness of aPN.** Two examined the cost-effectiveness of aPN [47,89], and three studies analyzed logistic and human resource costs [50,51,90]. In Kenya, aPN implementation ranged from $823 to $1619 per DALY averted, with greater cost-effectiveness under task-shifting scenarios, and a five-year implementation cost that remained below Kenya's per capita GDP [49]. In Mozambique, the cost per infection averted per year was $1,813 [90]. Operational costs in Kenya were mainly personnel (49%) and transport (13%) [50], while Mozambique's costs were primarily human resources (52%), testing kits (28%) and supplies (8%), with $5.82 per individual tested and $65.32 per new HIV diagnosis [90]. A community-based aPN incurred slightly higher costs than a facility-based model in Kenya [51].

## Qualitative findings

Of the total 76 studies reviewed, 13 are qualitative, comprising four on HIVST [36,40,69,108] and 11 on aPN [47,48,53,59,60,79,82,98,102]. Additionally, there are six mixed-method studies, with two focusing on HIVST [73,95] and four on aPN [46,78,100,106].

**Synthesized findings 1: HIVST gains high acceptance and feasibility due to its convenience, privacy, and cost-efficiency, supported by optimal timing, user competence, and targeted training for community health workers.** Ten individual findings from five studies were included, with eight unequivocal with illustrations, and two unequivocal without illustrations, as presented in supplementary S3 Table. It was reported that there is a willingness to distribute HIVST by married index cases to their partners [73]. Female PLHIV desired HIVST to test their male partners who previously refused facility-based testing due to privacy and time constraints [73]. Relationship intimacy, financial support, relationship commitment, and facilitated sexual risk discussion motivated FSWs to test partners [36]. Despite privacy concerns, male partners favoured home-based HIVST for its accessibility, convenience, and cost-saving [69,73]. Most index cases were confident about introducing HIVST to partners, with women believing they could demonstrate the HIVST kits without additional counselling [73]. Health workers in Malawi and Zambia favoured a choice-based approach but were concerned about effectiveness, time, and costs [69]. Community health workers (CHWs) expressed the training

needs to facilitate PNS using HIVST among MSM [47,112]. Both male and female index cases found HIVST more convenient than facility-based testing [73].

**Synthesized finding 2: HIVST fosters strategic communication and planning for partner testing, enhancing health safety, mutual disclosure, and supportive discussions on sexual health risks.** Eight individual findings from four studies were included, with seven unequivocal with illustrations, and one unequivocal without an illustration presented in supplementary S3 Table. Primary partners were convinced to use HIVST by FSWs through joint initiation and discussion, including withholding of sex. Even with the negative HIV results of their clients, FSWs insisted on using HIVST before unprotected sex to safeguard their health [36]. The benefits of HIVST for mutual status awareness, disclosure, and support, with careful communication to prevent confrontation, were recognized by both the index cases and their partners [40]. Stakeholders stressed the importance of prior communication by women to avoid surprises [69]. Individual PLHIV suggested bedtime as suitable for discussing HIVST with their partners [73].

**Synthesized finding 3: Effective HIVST index testing to partner notification, and partner testing is facilitated by streamlined policies and supportive environments but is often hindered by significant relational and societal barriers such as mistrust, stigma, gender norms, and safety concerns.** Eighteen individual findings from ten studies were included, comprising 14 unequivocal with illustrations, and four unequivocal without illustrations, as presented in supplementary S3 Table. FSWs cited suspicions of promiscuity and partner reluctance to test as challenges in HIVST partner notification [36]. Additional challenges included perceived partner notification ineffectiveness due to partners' HIV status and stigma, especially if partners were HIV-negative [69]. In West Africa, non-disclosure hindered couples counselling and secondary distribution [108]. Health professional reluctance, PLHIV hesitancy, and lack of disclosure support were obstacles to the index HIVST [108]. Potential challenges include trust issues, IPV risks, and harmful gender norms like male dominance [69,73,82,106]. Distrust led couples to question the index HIVST [73].

Among MSM, non-disclosure of HIV status was due to fear of revealing sexual orientation and lack of understanding [106]. In Indonesia, women saw partner reluctance due to fear of already being infected, rejection, and conflicting wishes to have children [47,78]. Incomplete partner information, safety concerns, and verbal abuse were reported as barriers in Kenya [46]. Fearing neighbours' awareness due to CHWs' visits, not being able to interpret results, and poor linkage to care were some concerns shared by PLHIV cases for the index HIVST [69].

An Indian study emphasized uniform policy and decentralizing HIVST partner testing, direct access, community participation, travel compensation, and positive mindsets among service providers [106]. In Indonesia, same-day couple testing eased partner testing, with HIV-negative husbands supporting wives' testing. Health professionals called for a uniform protocol and legal power for index testing [47,78]. Despite difficulties, participants aimed to reveal their status to primary partners, considering it as liberating and promoting harmony, clinic attendance and treatment adherence [98]. A mixed-method study showed that most index cases experienced less IPV when introducing the HIVST [73].

**Synthesized finding 4: Partner notification strategies prioritize assisted and choice-based referrals to enhance safety and effectiveness, addressing the limitations associated with passive methods, which, while cost-effective, may pose risks to client safety and privacy.** Nine individuals from three studies were included, with seven unequivocal with illustrations, and two unequivocal without illustrations, as presented in the supplementary S3 Table. In Malawi, health care providers and index cases supported a choice-based approach with home-based HIVST for HIV-negative women and index HIVST kits distribution for HIV-positive women. Invitation letters or calls by the healthcare provider effectively overcame partner rejection for HIVST [69]. In Kenya, some preferred couples testing or personal disclosure for aPN [48].

In Uganda, provider referral followed by contact referral was most preferred for partner testing [82]. Concerns of IPV and social harm were noted with pPR due to the non-involvement of healthcare providers in status disclosure. A modified pPR method, where couples initiate testing before status disclosure, was recommended as a safe, convenient, and cost-effective option, especially for married individuals needing more time to process their HIV-positive result [82]. The

findings related to stigma, IPV, gender norms, and provider constraints indicate the presence of contextual and policy-related factors influencing implementation.

## Integration of quantitative and qualitative evidence

Table 2 presents the Summary of the integration of quantitative and qualitative systematic review findings. Qualitative evidence highlighted the importance of privacy, commitment, and intimacy as key drivers for HIVST effectiveness. Quantitative data provided numerical insights, while qualitative data explained motivations and challenges, enriching the understanding of the HIVST and PNS index. The findings were skewed towards married couples, with limited data available on unmarried and undiagnosed key populations.

Quantitative data indicated that aPN elicit index partners for more new case detection than facility-based testing. Additionally, quantitative data provided various options for partner referrals, while qualitative data provided insights into social harms and barriers. The effectiveness and limitations of partner notification methods were highlighted by the data from both quantitative and qualitative sources. The qualitative data also emphasized the importance of HIVST decentralization and partner testing, and creating an enabling environment for index testing and notification.

## Discussion

Seventy-six studies were reviewed, resulting in the first quantitative and qualitative synthesis of the acceptability and feasibility of index HIVST for PNS in low to upper-middle-income countries, with a higher concentration of evidence from African countries and none from South Asia, except India, and only a few from East Asia. This highlights the need for further exploration to understand the context and setting of these geographical areas to effectively scale up HIVST for index testing, partner notification, and partner testing.

A higher uptake of HIVST than facility-based testing among ANC women was revealed by quantitative studies, who preferred delivering HIVST to their primary male partners. This aligns with qualitative findings where married women are willing to provide HIVST to partners who previously refused facility-based testing due to privacy and time constraints [73]. Financial incentives didn't significantly increase HIVST uptake among ANC women, suggesting non-cash incentives may be more sustainable. Non-disclosure of HIV status, healthcare providers' unwillingness, no trust, and social and gender harms were key barriers to HIVST and partner testing [69,73,82,106]. Considerations are needed to evaluate the index HIVST and partner testing among unmarried individuals, especially PLHIV living on antiretroviral therapy (ART) who have not disclosed their status. Previous research also shows that the index HIVST enhanced partner testing and ART initiation without added risk [70]. Similarly, a meta-analysis also revealed that assisted HIVST or secondary distribution resulted in higher linkage to confirmatory testing and ART initiation [113].In addition, the review identified a significant gap in the literature regarding unmarried individuals and undiagnosed key populations, particularly those on ART who have not disclosed their HIV status. These groups are often underrepresented in studies exploring the feasibility and acceptability of index HIVST for partner notification [114]. Future research should prioritize understanding the unique barriers and needs of these populations to ensure inclusive and effective partner services through contextually appropriate methods in LMICs.

MSM index partners showed a high uptake of HIVST compared with facility-based testing [56,57]. Other research shows high acceptability and feasibility of HIVST among MSM due to privacy, convenience, desire to prevent HIV transmission, and sharing HIV status as a sign of love [103,105,115,116]. This high uptake may be linked to MSM's preventive behaviors. Understanding these behaviors is critical for developing targeted interventions to prevent HIV [59]. The evidence suggests that tailored HIV partner services, using implicit partner notification, and developing diverse anti-stigma interventions by addressing policy gaps are important for improving access to and coverage of these services [59].

This review highlights the need for effective logistics and operational strategies for sustainable index HIVST and partner testing. This is consistent with findings from other studies, which stress the effectiveness of HIVST with the wider availability of testing kits [117]. A previous meta-analysis found that HIVST is a safe and effective intervention among the general

**Table 2. Summary of the integration of quantitative and qualitative systematic review findings.**

| Components of review | Summary of key findings |
|---|---|
| Acceptability, feasibility and uptake of HIVST for partner notification compared with facility-based testing. | Quantitative Findings:<br>• Higher HIVST uptake vs. facility-based testing reported in 8 studies, with increases ranging from 14% to 70%.<br>Qualitative Findings:<br>• HIVST gains high acceptance and feasibility due to its convenience, privacy, and cost-efficiency, supported by optimal timing, user competence, and targeted training for community health workers.<br>• HIVST fosters strategic communication and planning for partner testing, enhancing health safety, mutual disclosure, and supportive discussions on sexual health risks.<br>• Effective HIVST index testing partner notification is facilitated by supportive policies and environment, but hindered by relational and societal barriers such as mistrust, stigma, gender norms, and safety concerns. |
| Effectiveness of aPN services. | Quantitative Findings:<br>• Assisted partner notification, identified and tested index partners, leading to a higher positivity rate and linkage to care and treatment despite low partner elicitation.<br>Qualitative Findings:<br>• Barriers such as sexual and verbal violence, gender power imbalance, and fear of HIV status disclosure as the sole reasons for low elicitation.<br>• Differences in skills and knowledge of healthcare providers may have impacted effective elicitation. |
| Partner notification strategies with HIVST. | Quantitative findings<br>• Out of 12 studies, 4 RCTs showed that aPN and pPR effectively distributed HIVST kits, with testing rates of 35% to 71%.<br>• Seven other studies used varied methods like ANC-based, peer-led, digital, or provider-assisted models, with partner testing rates from 28.2% to 94.9%.<br>Qualitative findings<br>• Partner notification strategies prioritize assisted and choice-based referrals to enhance safety and effectiveness, addressing the limitations associated with passive methods, which, while cost-effective, may pose risks to client safety and privacy. |
| Partner notification strategies without HIVST. | Quantitative Findings:<br>• Partner notification preferences vary across index cases and key populations, influenced by context and method.<br>• Five studies showed a higher uptake of provider referral (aPN) than pPR<br>• Other strategies, like contact-based and self-disclosure, also featured in the literature but showed more variable outcomes depending on the setting.<br>Qualitative Findings:<br>• Partner notification strategies prioritize assisted and choice-based referrals for safety and effectiveness, addressing the limitations and risks to clients' safety and privacy associated with passive methods. |
| Barriers, facilitators, and social harms associated with HIVST for partner testing. | Quantitative Findings:<br>• Incidence of social harm (relationship dissolution, intimate partner violence, verbal abuse, and loss of finances) was reported about the secondary distribution of HIVST for index testing, but at a low prevalence.<br>Qualitative Findings:<br>• Possible risks associated with passive referral, such as intimate partner violence and self-harm upon disclosure in the absence of healthcare providers' mediation.<br>• Possible risks associated with passive referral are due to stigmatization and discrimination associated with gender identity, sexual orientation and the criminalization of sex work and injecting drug use. |
| Cost-effectiveness of aPN. | Quantitative findings<br>• Despite the operational and logistic costs, aPN interventions have reported low costs per infection averted, highlighting their cost-effectiveness<br>• The main cost drivers in Kenya (49%) are personal and (13%) transport. In Mozambique (52%), human resource test kit (28%)<br>• Community-based aPN is slightly more expensive than facility-based aPN in Kenya.<br>Qualitative findings<br>• Despite privacy concerns, male partners favored home-based HIVST for its accessibility, convenience, and cost-saving<br>• Health workers in Malawi and Zambia favoured a choice-based approach but were concerned about effectiveness, time, and costs. |

population with a range of delivery models, identifying and linking additional people with HIV to care [117]. Further, the previous research also showed that effective partner notification for HIV + MSM through CBO collaboration is promising [58]. However, there is limited information on the acceptability and feasibility of HIVST partner notification among MSM with lower socioeconomic status or those who have not disclosed their HIV status and sexual orientation, indicating the need for further research. Engaging key population subgroups is important to enhance HIVST uptake [118]. The review emphasizes the importance of integrating HIVST into community-based index testing and aPN, including enhancing the core competency of key service providers in index HIVST and partner notification for wider reach [58].

Relationship commitment and financial support were reported as motivating factors for high HIVST partner testing by FSWs [36,85]. Further, the previous review indicated that discussions on sexual risk and prevention were facilitated by HIVST, leading to partner testing among FSWs [109]. This suggests that HIVST promotes serostatus awareness and risk reduction among FSWs and their partners. However, negative reactions such as relationship strain, emotional distress, or hesitation to test indicate the need for support services and partner awareness [118]. Alternative HIVST delivery methods, like direct delivery or partner referral, may reduce sexual behavior risk [85]. Additional studies are needed to understand the impact of HIVST on sexual behaviors and decision-making. Investigating clients' perceptions of HIVST kit distribution by FSWs is crucial, despite FSWs' commitment to using HIVST for partner testing.

Despite low partner elicitation ratios among aPN-identified index partners, this case often results in a higher positivity rate and treatment coverage, highlighting the targeted effectiveness of aPN approaches [41,45,58,62–66,68,80,81,85–88,91,92,104,109]. This is consistent with qualitative evidence where low partner elicitation was due to barriers like sexual and verbal violence, gender power imbalance, and fear of HIV status [57,69,70,73,82,97,106,111] along with differences in health workers' skills [116]. The previous review stressed the importance of a balanced and context-specific approach to improve HIV partner notification among key populations in LMICs by harnessing facilitators alongside addressing barriers [109]. High testing uptake and increased case diagnosis suggest aPN as a viable alternative to index HIVST. This aligns with previous findings where aPN was found to be effective in case diagnosis, safe for partner notification with minimal social harms, and thus feasible for expansion with close monitoring of adverse events [105]. Furthermore, evidence elsewhere highlighted that targeted community-based index testing and aPN are cost-effective [90], which is consistent with your finding. The evidence outside this review also showed that task shifting of HIV testing, focusing on the targeted population, is a cost-effective and sustainable approach [117,118]. This supports the feasibility and acceptability of aPN universally, prioritizing male partners to bridge the HIV case detection and gender gap. However, this review could not determine the cost-effectiveness of index HIVST due to the lack of specific studies during the review period, highlighting a need for further research in this area.

Prior research has found that aPN is effective in case detection but has lower linkages to care compared to community-based testing [87]. Therefore, programs should strengthen the support system for HIV disclosure and consider engaging non-medical staff in HIVST distribution to involve partners in the HIV continuum of care. This aligns with studies showing the effectiveness of integrating aPN and HIVST within community-based testing to reach untested and undiagnosed key populations [87,107]. The review highlights that sustainable aPN and index HIVST is a feasible strategy to support HIV and AIDS epidemic control. However, no quantitative study assessing the effectiveness of pPR for index testing to partner notification was found, but its effectiveness as one of the approaches for partner notification has been reported by several studies, as discussed below [68,82,84–86,109].

Partner notification strategies are not one-size-fits-all and must be tailored to population-specific preferences and context. Our review indicates that preferences for partner notification vary. For instance, key populations of MSM, FSWs and PWID in Vietnam preferred provider referral due to the risk associated with pPR [107], linked to concerns about social harms without healthcare providers' mediation [82]. However, in Zambia and Vietnam, key populations preferred pPR over other assisted partner notification [68,85], driven by stigmatization related to gender identity and sexual orientation [78]. The previous findings also revealed that MSM fear disclosing their sexual orientation to key service providers [78].

Female index cases favoured pPR for primary partners [82,84,86,109] and provider referral for casual partners [69,82]. This is consistent with qualitative findings where female index cases chose HIVST for primary partners due to feelings of commitment, financial support, intimacy, and partner hesitancy to visit health centres [36,107]. Safety and confidentiality concerns led women to prefer contact referrals for casual partners. The previous research also showed that contact referrals were suitable for couples needing more time to process HIV-positive results [69,82]. A choice-based HIVST approach was preferred, with HIV-positive individuals choosing direct HIVST distribution and HIV-negative individuals preferring home-based HIVST through trained providers. Although different approaches for partner HIVST are accepted, implementing challenges exist in choice-based methods [69]. Prior studies also revealed that stigma and discrimination are the key barriers for men and women in accessing HIV testing and care. Evidence shows that these differences across genders in accessing services are influenced by sociocultural norms [119]. For example, women prioritize dyadic harmony and personal safety in HIV disclosure to mitigate risks of violence and discrimination [120,121]. While the masculinity expectations and privacy concerns of the men discourage them from seeking facility-based testing in many low- and middle-income settings [120–122]. This shows that adapting index HIVST strategies to diverse preferences and needs while addressing implementation challenges is important.

Index cases and healthcare providers preferred aPN for its safety over the cost-effective but socially harmful pPR. Therefore, a modified pPR encourages index cases to initiate partner testing before disclosing their status, especially for married individuals, and is considered suitable [82]. Incomplete information and social harm are key challenges healthcare providers highlight for index testing in a qualitative study [46]. Evidence outside this review also revealed that pPR is inexpensive and does not require additional staffing or costs. However, it is associated with a higher likelihood of IPV [116]. Stakeholders supported aPN, particularly dual referral methods [79]. This highlights unique challenges and opportunities in aPN and HIVST for healthcare workers. This review suggests the need for a uniform policy, protocol, and legal power to carry out index testing [78]. The WHO and UNAIDS policy guidelines highlight that differences in national legal frameworks, informed consent requirements, and the criminalization of key populations in many LMICs continue to undermine the HIV response, including feasibility, scalability, and integration of aPN and HIVST into existing partner notification services [10,121–123].

Tailored services with varied aPN options and context-based protocols and guidelines are important to address confidentiality and barriers, demanding flexible aPN options tailored to relationship dynamics. The evidence suggests that diverse notification options facilitate HIV disclosure, leading to timely testing and care [82]. This review found that pPR and interaction with health facilities may influence the consistency and quality of aPN implementation and partner-delivered HIVST. The evidence suggests that fidelity assessments are crucial for anticipating and mitigating contextual effects for scale-up index testing to PNS [46]. Differences in acceptability and feasibility vary by population, gender, and type of relationship. This shows the intersections of stigma, safety, and the legal environment among MSM and FSW, hindering disclosure and partner notification. For the women living with HIV, privacy and safety shaped their preferences for primary versus casual sexual partners. For PWID, legal and policy barriers inhibited their participation in index testing and partner notification. This highlights the importance of implementing rights-sensitive and context-specific HIVST and partner notification services, particularly in LMICs. Overall, these review findings are consistent with the current WHO and UNAIDS guidelines, underlining the importance of integrating HIVST for index testing within partner notification services to enhance case diagnosis and linkage to treatment.

The key strengths of this review include a robust search strategy and systematic approach, as well as the use of JBI convergent segregated methods to integrate both qualitative and quantitative findings in a transparent and rigorous manner. The meta-aggregation process further minimized the risk of interpretation bias. Notably, this is the first review to assess the effectiveness, acceptability and feasibility of HIVST and aPN for partner notification, providing comparative insights across different populations, genders and settings within a single mixed-methods framework in low- to upper-middle-income countries.

There were some limitations. Studying heterogeneity hindered meta-analysis; unpublished or ongoing trials were excluded. Although GRADE or CERQual assessments were not applied, the methodological quality and bias were assessed using the MetaQAT framework. This review was limited to studies published in English, and while no non-English studies were encountered during screening, relevant studies in other languages may have been missed. The search strategy was limited to four databases, which may have missed some relevant studies from LMICs published in sources such as LILACS or African Index Medicus. No grey literature was reviewed, and we may have missed studies reported outside of the peer-reviewed literature. However, the reference lists of the selected articles were examined for additional eligible articles. Furthermore, the search did not include the term "secondary distribution", which may have resulted in the exclusion of some studies related to HIVST distribution by key populations to their partners. All studies were included regardless of quality; those with stronger methodologies contributed more substantially to the findings on the effectiveness of index HIVST and aPN. Focusing on low- to upper-middle-income countries, excluded high-income countries. The 2016 cut-off date may have excluded studies related to conventional aPN.

## Conclusions

HIVST for index testing to partner notification is acceptable and feasible due to its convenience, privacy, and cost-efficiency, supported by optimal timing, user competence, and targeted training for community health workers. HIVST enhances strategic communication and planning for partner testing, promoting health safety, mutual disclosure, and supportive discussions on sexual health risks. Effective index HIVST to partner notification and partner testing are facilitated by streamlined policies and supportive environments, but are often hindered by relational and societal barriers like mistrust, stigma, gender norms, and safety concerns. Despite low partner elicitation rations among aPN-identified index cases, this approach often results in a higher positivity rate and treatment coverage, highlighting the targeted effectiveness of aPN. This review found no evidence of cost-effectiveness for index HIVST; however, community-based index testing and aPN were identified as cost-effective strategies. It is also important to note a critical research gap concerning unmarried and undiagnosed key populations. This imbalance in evidence may hinder the generalizability of findings and perpetuate inequities in access to partner notification services.

Furthermore, most studies were concentrated in African countries, with limited representation from South Asia and East Asia, thereby creating a gap in contextual understanding. Future research should prioritize contextually appropriate methods in LMICs to overcome the sociocultural and structural barriers, including relational complexities among diverse populations. This review also reveals the intersection of stigma, safety and the legal environment among MSM, FSW and PWID populations, hindering disclosure and partner notification. Partner notification strategies should prioritize assisted and choice-based referrals to enhance safety and effectiveness, addressing the risk associated with pPR, which, while cost-effective, may compromise client safety and privacy. This review's strength lies in its robust search strategy, use of the JBI convergent segregated method, and meta-aggregation. However, the study heterogeneity limited the meta-analysis, and restricting to English may have excluded relevant non-English studies. Finally, this review demonstrates that integrating HIVST within partner notification services presents an effective alternative testing option for reaching untested and undiagnosed individuals, thereby contributing to the achievement of the UNAIDS 95-95-95 target in LMICs.

## Supporting information

**S1 Table. PRISMA checklist.**
(DOCX)

**S2 Table. Full search strategy.**
(DOCX)

**S3 Table. Qualitative meta-aggregation synthesized findings.**
(DOCX)

**S4 Table. Full-text studies with inclusion-exclusion reasons.**
(DOCX)

**S5 Table. Characteristics of included studies.**
(DOCX)

**S6 Table. Quality appraisal result and risk of bias.**
(DOCX)

**S7 Table. Article search results for Web of Science database.**
(DOCX)

## Acknowledgments

The authors have no acknowledgements to declare.

## Author contributions

**Conceptualization:** Lekey Khandu.

**Formal analysis:** Lekey Khandu.

**Investigation:** Lekey Khandu.

**Methodology:** Lekey Khandu.

**Software:** Lekey Khandu.

**Supervision:** Jonathan Hallett, Gemma Crawford, Justine E Leavy, Daniel Vujcich.

**Validation:** Jonathan Hallett, Gemma Crawford, Justine E Leavy.

**Writing – original draft:** Lekey Khandu.

**Writing – review & editing:** Jonathan Hallett, Gemma Crawford, Justine E Leavy, Daniel Vujcich.

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
