## [Decision Letter · Decision Letter 0]

15 May 2025

Dear Dr. Khandu,

Thank you for submitting your manuscript to PLOS ONE. After careful consideration, we feel that it has merit but does not fully meet PLOS ONE’s publication criteria as it currently stands. Therefore, we invite you to submit a revised version of the manuscript that addresses the points raised during the review process.

We look forward to receiving your revised manuscript.

Kind regards,

Mulualem Endeshaw

Academic Editor

PLOS ONE

 [PhD Scholarship]. 

[This study was part of my doctoral research program, supported by Curtin University, Western Australia. As a PhD student from Bhutan, a low- to middle-income country, I received this support to pursue my studies. The author declares that there is no competing interests for this study.]

[PhD Scholarship]. 

5. As required by our policy on Data Availability, please ensure your manuscript or supplementary information includes the following:

Reviewers' comments:

Reviewer's Responses to Questions

**Comments to the Author**

1. Is the manuscript technically sound, and do the data support the conclusions?

Reviewer #1: Partly

2. Has the statistical analysis been performed appropriately and rigorously?

Reviewer #1: N/A

3. Have the authors made all data underlying the findings in their manuscript fully available?

Reviewer #1: Yes

4. Is the manuscript presented in an intelligible fashion and written in standard English?

Reviewer #1: No

Reviewer #1: 1. Major Comments

1.1. Title: The current title is excessively long. I suggest revising it to: "Acceptability and feasibility of HIV self-testing for partner notification in low- to upper-middle-income countries: A mixed-methods systematic review."

1.2. Objective and Scope:

• Line 101: The connection between question 2 and HIV self-testing (HIVST) as announced in the title is unclear. This link should be clarified, or the objective should be omitted. The overall objective, as stated in the discussion (e.g., lines 419 or 530-531), focuses on assessing the acceptability and feasibility of HIVST for partner notification (PN). Including question 2 dilutes the focus of the paper.

• Line 263: Some results suggest that pPR (partner-provided referral) is also related to HIVST. Why dedicate one question solely to assisted partner notification (aPN) without addressing pPR? This inconsistency should be resolved. See my previous comment.

• Introduction: The overall objective of the paper should be explicitly stated in the introduction for clarity.

1.3. Previous Literature:

• Line 114-116: Be specific about what is novel in your review. Other systematic reviews have already covered some aspects, including the acceptability and use of HIVST (e.g., link 1, link 2) and feasibility (e.g., link 3). Highlight explicitly how your work differs or complements these studies.

1.4. Study Methods:

• Line 122-125:

o Consider mentioning how you treated mixed-method studies explicitly.

o The statement "The qualitative components examined barriers and facilitators for index testing for PNS" raises questions. Is this focus on qualitative research due to a lack of quantitative studies? The JBI mixed-method systematic review manual (e.g., JBI guide) shows that barriers and enablers can also be addressed quantitatively. The authors should explain the divide between qualitative and quantitative approaches for each component of the research.

1.5. Search Strategy:

• Line 129: The term "secondary distribution" is commonly used in studies on key populations for partner testing. Excluding this term from the search may have caused you to miss relevant publications, such as those from the ATLAS project (link to ATLAS publications) which focus on distribution to partners of both diagnosed and undiagnosed key populations.

1.6. Heterogeneity:

• Line 161: Clarify the sources of heterogeneity identified.

• Line 217: Author state that no sensitivity analyses or investigations into heterogeneity were conducted, yet heterogeneity is mentioned in line 161. This contradiction should be resolved.

1.7. Subsection Relevance:

• Line 273: The relevance of the section “Methods for index testing to PN” is unclear. Given Question 1, studies comparing HIVST to facility-based methods should be analyzed together.

• Line 307-316: This section’s relevance to the research questions is unclear and could be omitted.

1.8. Table 2: The table should be restructured to align with the three research questions of the paper.

1.9. Language and Statements:

• Line 527: The authors state that "non-English languages were considered," but only English terms were used in the search. This statement is misleading and should be removed.

• Line 528-529: Clarify how this aspect of the methodology was implemented.

2. Minor Comments

2.1. Line 75-77: Add a transition sentence to improve the connection between the two paragraphs.

2.2. Abbreviations: Reduce the number of abbreviations to improve readability, keeping only those that are frequently used in the text.

2.3. Authorship Details: In the methods section, it is unusual to specify which author performed which tasks. Instead, mention the number of authors involved in each task when relevant, without using initials.

2.4. Line 242: This sentence is unclear and should be rewritten for better comprehension.

2.5. Line 433: The term "ART cases" is ambiguous. Define "ART" at its first use and clarify its meaning in this context.

2.6. Line 520: The phrase "the fidelity of aPN" is unclear and should be rephrased or elaborated.

2.7. Line 522: The "Strengths and Limitations" subsection appears to be the only subsection within the discussion. Consider removing the subsection title and integrating the content into the main discussion.

2.8. Line 525: The statement about "inadequate information among non-married and undiagnosed key populations" is vague. If this is a gap in the literature rather than a limitation of the current study, it should not be listed as a limitation of the paper.

**Do you want your identity to be public for this peer review?** For information about this choice, including consent withdrawal, please see our Privacy Policy

Reviewer #1: No

---

## [Author Response · Author response to Decision Letter 1]

13 Jun 2025

We have submitted a separate file entitled "Response to Reviewer" along with the manuscript and other supporting documents as an attachment for your kind perusal.

---

## [Decision Letter · Decision Letter 1]

27 Oct 2025

Dear Dr. Khandu,

Thank you for submitting your manuscript to PLOS ONE. After careful consideration, we feel that it has merit but does not fully meet PLOS ONE’s publication criteria as it currently stands. Therefore, we invite you to submit a revised version of the manuscript that addresses the points raised during the review process.

**Your revised manuscript received one review.  The reviewer still identified several important issues in your work. Please consider the attached comments and provide point-by-point responses.  **

We look forward to receiving your revised manuscript.

Kind regards,

Yury E Khudyakov, PhD

Academic Editor

PLOS ONE

**Journal Requirements:**

Reviewers' comments:

Reviewer's Responses to Questions

**Comments to the Author**

Reviewer #2: (No Response)

2. Is the manuscript technically sound, and do the data support the conclusions?

Reviewer #2: Partly

3. Has the statistical analysis been performed appropriately and rigorously?

Reviewer #2: Yes

4. Have the authors made all data underlying the findings in their manuscript fully available?

Reviewer #2: Yes

5. Is the manuscript presented in an intelligible fashion and written in standard English?

Reviewer #2: Yes

**Reviewer #2:**  The manuscript addresses an important and timely topic—the acceptability and feasibility of HIV self-testing (HIVST) for partner notification among index cases and key populations in low- to upper-middle-income countries. The focus is highly relevant to advancing HIV testing services, reaching underserved groups, and informing strategies toward the UNAIDS 95-95-95 targets. The mixed-methods approach is appropriate, given the need to capture both quantitative outcomes (uptake, linkage, yield) and qualitative insights (barriers, facilitators, acceptability), and the inclusion of diverse key populations adds strength. The review is systematic, draws on multiple databases, and presents a broad overview of available evidence, with clear implications for addressing case detection gaps.

That said, the manuscript in its current form requires strengthening before publication. The Introduction provides useful background but does not clearly articulate the specific evidence gaps, rationale for using a mixed-methods review, or the global policy significance of this work. The Methods section lacks detail on protocol registration, search strategy transparency, study selection procedures, outcome definitions, risk-of-bias tools, and certainty assessment, all of which are essential for reproducibility and rigor in systematic reviews. The Results are comprehensive but tend to be descriptive and would benefit from a clearer structure aligned with the stated objectives, more explicit subgroup analyses (e.g., by population, region, gender), and better integration of quantitative and qualitative findings. The Discussion reiterates findings but offers limited critical interpretation; it underplays the equity, policy, and implementation context and does not fully engage with global targets or comparative literature. The Conclusion is concise but overly descriptive, lacking interpretive strength, programmatic implications, or a forward-looking research agenda.

Overall, this review contributes important evidence on HIVST in the context of partner notification, with potential value for policymakers, implementers, and researchers. However, revisions are needed to improve methodological transparency, strengthen critical interpretation, and situate findings within the broader global health context. By clarifying evidence gaps, integrating equity considerations, explicitly mapping findings to policy and practice, and offering a sharper research agenda, the manuscript can achieve greater clarity, rigor, and impact.

**Do you want your identity to be public for this peer review?** For information about this choice, including consent withdrawal, please see our Privacy Policy

Reviewer #2: **Yes: ** Laufred I. Hernandez

---

## [Author Response · Author response to Decision Letter 2]

18 Nov 2025

I have a uploaded a separate Response to Reviewer file.

---

## [Editor Report · Decision Letter 2]

25 Nov 2025

HIV self-testing and partner notification strategies for key populations in low- to upper-middle-income countries: A mixed-methods systematic review

PONE-D-24-43591R2

Dear Dr. Khandu,

We’re pleased to inform you that your manuscript has been judged scientifically suitable for publication and will be formally accepted for publication once it meets all outstanding technical requirements.

Kind regards,

Yury E Khudyakov, PhD

Academic Editor

PLOS ONE
---

## [Editor Report · Acceptance letter]

PONE-D-24-43591R2

PLOS One

Dear Dr. Khandu,

I'm pleased to inform you that your manuscript has been deemed suitable for publication in PLOS One. Congratulations! Your manuscript is now being handed over to our production team.

Kind regards,

on behalf of

Dr. Yury E Khudyakov

Academic Editor

PLOS One